# Confirmation of ‘Pollen- and Seed-Specific Gene Deletor’ System Efficiency for Transgene Excision from Transgenic *Nicotiana tabacum* under Field Conditions

**DOI:** 10.3390/ijms24021160

**Published:** 2023-01-06

**Authors:** Zhenzhen Duan, Mingyang He, Sehrish Akbar, Degang Zhao, Muqing Zhang, Yi Li, Wei Yao

**Affiliations:** 1State Key Laboratory for Conservation and Utilization of Subtropical Agro-Biological Resources & Guangxi Key Laboratory for Sugarcane Biology, Guangxi University, Nanning 530004, China; 2Department of Plant Science and Landscape Architecture, University of Connecticut, Storrs, CT 06269, USA; 3Plant Conservation Technology Center, Guizhou Key Laboratory of Agricultural Biotechnology, Guizhou Academy of Agricultural Sciences, Guiyang 550006, China

**Keywords:** auto transgene excision, FLP recombinase, field evaluation, transgene flow, Gene Deletor

## Abstract

The commercial application of genetically modified plants has been seriously impeded by public concern surrounding the potential risks posed by such plants to the ecosystem and human health. Previously, we have developed a ‘pollen- and seed-specific Gene Deletor’ system that automatically excised all transgenes from the pollen and seeds of greenhouse-grown transgenic *Nicotiana tabacum*. In this study, we conducted seven field experiments over three consecutive years to evaluate the stability of transgene excision under field conditions. Our results showed that transgenes were stably excised from transgenic *Nicotiana tabacum* under field conditions with 100% efficiency. The stability of transgene excision was confirmed based on PCR, as well as the GUS staining patterns of various organs (roots, leaves, petiole, stem, flower, fruit, and seeds) from transgenic *N. tabacum*. In six transgenic lines (D4, D10, D31, D56, and D43), the transgenes were stably deleted in the T0 and T1 generations. Thus, the ‘Gene Deletor’ system is an efficient and reliable method to reduce pollen- and seed-mediated unintentional gene flow. This system might help to alleviate the food safety concerns associated with transgenic crops.

## 1. Introduction

Sexual crossbreeding has been influential in the genetic improvement of crop species [1]. However, crossbreeding is a time-consuming, laborious process that is limited by reproductive isolation. In contrast, plant gene transfer introduces functional genes directly into host plants, thus eliminating the requirement for reproductive compatibility and allowing rapid directional selection [2]. Thus, plant gene transfer and other genetic modification (GM) techniques are powerful and efficient methods of plant genetic improvement. To date, many important commercial crops have been subjected to GM, including corn, soybean, canola, and cotton [3]. These GM crops are higher in yield and more cost effective than natural crops, as they require less fertilizer, herbicides, labor, and energy [3]. Since the first commercial use of transgenic crops in 1996, the global area devoted to transgenic crop growth has increased ~112-fold, reaching 190.4 million hectares in 2019 (http://www.isaaa.org/resources/publications/pocketk/16/default.asp (accessed on 3 January 2022)).

Scientists, environmentalists, politicians, and the public have suggested that transgenes (e.g., genes resistant to bacteria, herbicides, stress, viruses, insects, and diseases) and transgene products might pose health risks to humans and other organisms [4,5,6,7]. It has also been proposed that transgenes might escape from GM crops and integrate into the genomes of non-transgenic varieties, wild species, and crop-related species that are considered weeds [8,9]. This might lead to the contamination of non-transgenic and wild genomes as well as the generation of super or/and invasive weeds [10]. These concerns have severely impeded the development and use of GM crops [10].

Molecular evidence for transgene flow via pollen has been documented in native Mexican maize *(Zea mays*) [11], two different types of transgenic *Brassica napus* [12,13], and creeping bentgrass (*Agrostis stolonifera*) [14]. This suggests that the use of transgenic crops may carry a risk of transgene introgression [10,15]. Alternatively, transgenic seed dispersal via spillage during harvesting, transportation, and handling may also lead to inadvertent transgene flow [16,17]. Finally, transgenic seeds, left in fields accidentally, may germinate and grow together with non-transgenic crops, leading to the genetic contamination of the non-transgenic crops [18,19]. Thus, the possibility of unintentional mixtures of transgenic and non-transgenic genomes has led to social concerns about contamination, resulting in considerable economic losses for bio-tech companies [20]. Therefore, the prevention of transgene flow, whether via pollen or seeds, is critical for the safe and socially acceptable use of transgenic crops.

Several strategies have been proposed to eliminate or suppress unintentional transgene escape from transgenic plant populations [21]. Some of these strategies, such as transgenic mitigation, male sterility, chloroplast engineering, maternal inheritance, and transgene excision, are more practical [22]. Transgene dispersal tends to occur through the movement of pollen and seeds; therefore, biocontainment strategies targeting pollen and seeds have attracted considerable attention [10,15,23]. Site-specific recombinase-mediated transgene excision is considered the most promising transgene biocontainment strategy [24]. This system is comprised of recombinase Cre, R, or FLP, along with two similar or identical palindromic recognition sites (lox, RS, or FRT). The recognition site comprises an inverted repeat sequence, which binds recombinase, and a short asymmetric spacer sequence, which facilitates the exchange of DNA strands. If the DNA molecule contains two recognition sites in the cis position, the DNA segment will be excised if it is flanked by two directionally oriented sites and will be inverted if it is flanked by two oppositely oriented sites. If the DNA molecule contains two recognition sites in trans, reciprocal translocation between two DNA molecules is possible; alternatively, the two DNA molecules will become integrated if one of them is circular [25,26]. Several site-specific recombinase-mediated transgene excision systems, including Cre-loxRP, FLP-FT, and R-RS, have been tested in transgenic tobacco, *Arabidopsis*, strawberry, tomato, soybean, maize, and banana [27,28]. However, the transgene excision efficacies of “Gene Deletor” systems in transgenic plants have yet to be tested under field conditions.

Previously, we reported that in the greenhouse, Cre- or FLP-recombinase specifically deleted transgenes from pollen and/or seeds, regulated by a pollen- and/or seed-specific promoter [29]. However, system efficacy was only demonstrated in transgenic lines in the greenhouse. It was unclear whether this system would also function efficiently under field conditions. Therefore, in this study, we extended our previous work to determine whether the ‘Gene Deletor’ system also successfully deleted transgenes from transgenic *Nicotiana tabacum* under field conditions. In the present study, we selected transgenic *Nicotiana tabacum* (carrying the ‘pollen- and seed-specific Gene Deletor’ cassette) with 100% transgene excision efficiency in the greenhouse. Then, we tested the stability of transgene excision in the field over three consecutive years and demonstrated the 100% efficacy of transgene excision from pollen and seeds.

## 2. Results

### 2.1. Gene Deletor Vector Construction

A Gene Deletor vector was constructed to enhance the deletion efficiency of transgene from the targeted tissues. We constructed the Gene Deletor vector by fusing gene sequences from FLP/FRT and CRE/loxP systems and cloning these sequences into the pBIN19 binary vector (“LF” is an abbreviation for the loxP-FRT fusion sequence). The Gene Deletor vector contained the *NPTII* kanamycin resistance gene, which served as a selectable marker gene. The *GUS* reporter gene signaled the presence or absence of transgene expression, and *FLP* recombinase (PAB5) catalyzed site-specific DNA recombination, leading to the excision of the DNA between the two LoxP-FRT sites. The control vector lacked the PAB5 gene for site-specific recombination but contained the GUS::NPTII gene to indicate GUS activity.

### 2.2. Production and Confirmation of N. tabacum Transgenic Lines

Control and Gene Deletor plasmids were transformed into *N. tabacum* plants using the *Agrobacterium*-mediated transformation protocol [30]. We generated 202 independent transgenic lines using the control and Gene Deletor plasmids (Table 1).

Transgenic and wild-type plants grown in the greenhouse exhibited similar phenotypes and growth patterns. Independent transgenic lines were selected for kanamycin resistance. Transgenic plants were screened based on a histochemical analysis of β-glucuronidase (GUS) activity in leaf and root tissues. 

### 2.3. The ‘Gene Deletor’ System Excised Transgenes from the Pollen and Seeds of Greenhouse-Grown N. Tabacum

T1 seedlings germinated from self-pollinated seeds were GUS-stained to determine the transgene excision efficiency of each transgenic line. If the ‘Gene Deletor’ system functions correctly in a transgenic plant, all DNA (including transgenes) between the two fused loxP-FRT sites should be completely deleted in the pollen and seeds. Therefore, all T1 self-pollinated seedlings should be non-transgenic and should not exhibit GUS activity.

More than 75% of the T1 seedlings of the transgenic plants transformed with the control vector were positive for GUS activity (i.e., exhibited blue staining), following Mendel’s law of segregation. This indicated that the transgenes were not excised in the pollen and seeds produced by the control plants, and that some of the control plants had multiple transgenic loci. In contrast, less than 70% of the T1 seedlings transformed with the ‘Gene Deletor’ vector exhibited GUS activity (Figure 1A–F) (Table 1). Thus, transgenes were excised from most of the pollen and seeds produced by some of the lines transformed with the ‘Gene Deletor’ vector.

Consistent with our previous experiment [29], some transgenic plants with 100% excision efficiency were produced by each transformation with the ‘Gene Deletor’ cassette. Across all three experiments, we identified five independent lines with 100% excision efficiency (D4, D10, D31, D56, and D43) (Table 2). By comparing qPCR-calculated relative expressions to our standard curve, we determined that each of these lines possessed a single copy of the transgene. To provide controls for these lines, we also used our standard curve to identify control lines (i.e., lines transformed with the control cassette) in each experiment possessing a single copy of the transgene: C1, C14, and C6 (Table 2).

We next verified the transgene deletion efficiencies of the transgenic lines D4, D10 (generated in Experiment 1), D31, D56 (generated in Experiment 2), and D43 (generated in Experiment 3) (in comparison to the control lines C1, C14, and C6). In the control plants, GUS staining indicated that about 75% of the self-pollinated seedlings and about 50% of the hybrid seedlings exhibited GUS activity (Table 2).

These ratios were consistent with Mendel’s law of segregation, suggesting that the transgenes were not deleted in these control plants. However, across the 5864 self-pollinated seedlings and the 4808 cross-pollinated seedlings produced by the transgenic lines (D4, D10, D31, D56, and D43), none were positive for GUS activity (Table 2). This indicated that the transgenes had been completely excised from the pollen and seeds of these lines.

### 2.4. The ‘Gene Deletor’ System Stably and Completely Excised Transgenes from the Pollen and Seeds Produced by Transgenic N. tabacum Grown in the Field

The stability and expression of the Gene Deletor system were also analyzed in field-grown plants. GUS staining, corresponding to GUS activity, was detected in the presence and its expression in the roots, leaves, petioles, stems, flower stalks, and flowers of both the ‘Gene Deletor’-transformed lines (D4, D10, D31, D56, and D43) and the control transgenic lines (C1, C14, and C6) grown in the field. Consistent with this, PCR analysis showed that the *FLP* gene was present in the genomes of all transgene-excised transgenic lines tested (lines D4, D10, D31, D56, and D43) but not in control lines (Appendix A). Additionally, GUS staining indicated positive GUS activity in the mature pollen grains, immature seeds, and immature fruits from the control plants (C1, C14, and C6) (Figure 2A,C,E), but not in those from the ‘Gene Deletor’-transformed plants (D4, D10, D31, D56, and D43) (Figure 2B,D,F). This demonstrated that the transgenes had been successfully excised from the pollen and seeds produced by field-grown plants transformed with the ‘Gene Deletor’ system.

Our field experiments, which included thousands of control seedlings and tens of thousands of ‘Gene Deletor’-transformed seedlings, showed that 75% of the self-pollinated T1 control seedlings showed GUS activity (Figure 2E), as did 50% of the hybrid control seedlings (Table 3). However, no GUS activity was observed in any self-pollinated T1 ‘Gene Deletor’-transformed seedlings (Figure 2F) nor in any hybrid ‘Gene Deletor’-transformed seedlings (Table 3). Thus, the transgenes were excised with 100% efficiency in the pollen and seeds produced by lines D4, D10, D31, D56, and D43, even when grown in the field under changeable climatic conditions. In contrast, transgenes in control plants (lines C1, C14, and C6) followed Mendel’s law of segregation.

### 2.5. Molecular Analysis of Transgene Excision from Pollen and Seeds

If the ‘Gene Deletor’ system functions correctly in a transgenic plant, the DNA between the two loxP-FRT sites, including the transgenes, should be excised, and only a short T-DNA residue, containing only a flanking sequence with an intact fused loxP-FRT sequence, should remain in the genome (Figure 3). Therefore, only the T-DNA residue, not the transgenes, should be PCR amplified in the seeds and pollen from the T1 progeny of a transgenic line (Figure 3). This implies that if PCR is performed with the primers LF-F and LF-R, a 7.6 kb fragment should be amplified from the T_0_ transgenic plants transformed with the ‘Gene Deletor’ vector, but a 0.2 kb fragment should be amplified from the T1 seedlings germinated from self- or cross-pollinated seeds.

Consistent with expectations, 7.6 kb fragments containing 35S-NPT::GUS-3′35S and PAB5-FLP-Tnos were PCR-amplified from the leaf tissues of the transgenic lines D4, D10, D31, D56, and D43 grown in the field, but 0.2 kb fragments (or no fragments) were PCR-amplified from each of their self-pollinated T1 seedlings (Appendix A). Specifically, of the 200–300 self-pollinated T1 seedlings analyzed per transgenic line, about 75% produced a 0.2 kb DNA fragment, while the rest did not produce any specific bands. None of the tested T1 seedlings produced a 7.6 kb fragment (Table 4).

Each 0.2 kb fragment had the same sequence: a short T-DNA residue containing an intact loxP-FRT fusion sequence. Thus, molecular analysis suggested that the transgenes had been completely excised from the pollen and seeds of the ‘Gene Deletor’-transformed transgenic plants grown in field.

## 3. Discussion

There has always been public concern regarding inadvertent foreign gene flow and the safety of transgenic foods. Preventing inadvertent gene flow out of transgenic crops is critical, because even very low levels of gene flow can permit the spread of highly adventitious genes. Furthermore, once an external gene becomes established, it is very difficult to prevent its spread [15,31]. Thus, it is important to develop automatic transgene excision systems that completely delete foreign genes from the pollen and seeds produced by transgenic plants. As climatic factors such as light, temperature, rainfall, and wind strongly affect plant growth and development, it is necessary to evaluate the stability and efficiency of any transgene deletion system under field conditions as well as under greenhouse conditions.

Here, we extended our previous study of the ‘Gene Deletor’ system [29] and showed that transgenic plants with 100% transgene excision efficiency were repeatably produced by transformation with the ‘Gene Deletor’ vector, in which *FLP* is controlled by the *Arabidopsis* PAB5 promoter. More importantly, we showed that these results were replicable under variable climatic conditions over three consecutive years, suggesting that the ‘Gene Deletor’ system is a reliable automatic transgene excision system. After transgene excision, only a very short foreign DNA fragment, containing the left- and right-border T-DNA sequences, as well as an intact loxP-FRT recognition sequence, remained in the genomes of the pollen and seeds produced by the transgenic plants. This short foreign DNA residue has no function and is not transcribed; therefore, these transgenic plants are environmentally friendly and confer no risk of inadvertent foreign gene flow via pollen and seeds. Thus, plants transformed with the ‘Gene Deletor’ vector possess the desired traits associated with transgenes but produce non-transgenic pollen and seeds due to automatic transgene excision. Previous studies reported the efficiency of a Cre/LoxP recombination system in both prokaryotes and eukaryotes. These studies showed the effectiveness of this system in the development of transgene free transgenic plants. [32] developed a FLP/LoxP-FRT recombinase system in *Escherichia coli*, where it was utilized as a gene switch to regulate the gene expression. Similarly, [33] generated selectable marker-free transgenic plants using a Cre/loxP recombination system, which was controlled by −46 minimal CaMV 35S promoter. T0 and T1 transgenic Arabidopsis plants showed marker-free plants harboring only excised constructs in their genomes. These reported studies have not highlighted the effect of any remaining foreign DNA fragments on the genome evolution of species. Consistently, our experimental analysis proved the efficient excision of transgenes without any adverse environmental and phenotypic effects.

The ‘Gene Deletor’ system may be especially useful for vegetatively propagating species. That is because vegetative propagation methods do not involve pollen or seeds, the transgene cannot escape during propagation, and the ‘Gene Deletor’ vector prevents transgene escape via subsequent seed and pollen production. This reduces the potential risk of negative environmental impacts due to inadvertent foreign gene flow. Previously, [34] described the use of LEAFY (*LFY*) controlled promoters to excise the PAB5 marker gene from bananas. The same strategy was implemented with Cavendish bananas, and the Gene Deletor vector driven by the *LFY* promoter excised 88.5% of the exogenous genes [28].

A modified version of the ‘Gene Deletor’ cassette may help to prevent foreign gene flow from the pollen and seeds of sexually propagated transgenic crops, such as soybeans, sunflowers, canola, corn, and sorghum, which tend to outcross to wild relatives [2,35]. The introduction of the chemically inducible *RNAi-FLP* gene into the ‘Gene Deletor’ cassette, in conjunction with the application of a non-phytotoxic induction agent, may repress FLP gene expression in pollen and seeds [29]. This would prevent the deletion of transgenes from the pollen and seeds and allow the production of certified seed stocks. Without the application of the induction agent, the subsequent generation of sexually produced plants would produce transgene-free pollen and seeds. This system might be especially useful for plants that are genetically modified for patenting purposes.

There are additional applications of the ‘Gene Deletor’ system. Firstly, non-transgenic seeds or plants could be produced from transgenic plants, allaying consumer concerns over transgenes in food. Secondly, the ‘Gene Deletor’ technology would allow farmers to replant viable but non-transgenic seeds harvested from transgenic plants. This is in contrast to the controversial ‘terminator seed’ technology [36], which generates transgenic plants that produce non-viable seeds. Finally, the ‘Gene Deletor’ system, when used in grain crops, eliminates the need to label and physically separate transgenic and non-transgenic grains after harvest.

## 4. Materials and Methods

### 4.1. Nicotiana tabacum L. Transformation Using the ‘Gene Deletor’ Vector

Transgenic *Nicotiana tabacum* plants were generated using the control vector or the “Gene Deletor” vector as described previously [29]. The previously generated “Gene Deletor” vector was comprised of two identical LoxP-FRT (termed “LF”) fusion sequences. In between the two LF sequences was located the GUS::NPTII fusion gene, driven by the constitutive CaMV 35S promoter, and the FLP yeast recombinase gene, controlled by the *Arabidopsis* pollen, ovule, and early embryo specific promoter PAB5 [37] (Appendix A). The control vector was identical to the “Gene Deletor” vector, except it was lacking the PAB5-FLP-NOS cassette (Appendix A). *Agrobacterium tumefacians* strain EHA105 was transformed with the “Gene Deletor” or control plasmid as described previously [30].

Wild-type tobacco plants were simultaneously generated in vitro (at 23 °C, with a 14 h light/10 h dark cycle and 50–60% humidity) and used as controls. Flower maturity was determined using the stages proposed by [38]; we modified this scale to include an extra stage (0) before stage 1 (Appendix A).

### 4.2. GUS Histochemical Staining

GUS staining of transgenic leaf and root tissues was performed by dipping the tissues in GUS staining solution (200 mM NaPO_4_ buffer (pH 7.0), 10 mM mercaptoethanol, 10 mM EDTA, 0.1% (*w/v*) sodium azide, 0.1% (*w/v*) Triton X-100, and 0.5 mg/mL X-Gluc) and incubating the tissues at 37 °C overnight. After incubation, leaf sections and roots were de-stained in 70% ethanol. The de-staining process was repeated four times prior to visual analysis.

### 4.3. PCR Detection of Transgenes

Genomic DNA was isolated from the leaves of the GUS-positive plants and purified as described previously [39]. Primer pairs were designed to detect pFLP and pGUS (pFLP-F (5′-TCAATTGTGGAAGATTCAGCGA-3′) and pFLP-R (5′-CCCTTGCGCTAAAGAAGTATATG-3′); pGUS-F (GTTACGTCCTGTAGAAACCCCAACC) and pGUS-R (CTGCCCAACCTTTCGGTATAAGAC)). The PCR cycling conditions were as follows: 98 °C for 5 min; 35 cycles of 98 °C for 5 s, 65 °C for 10 s, and 72 °C for 90 s; and a final extension at 72 °C for 3 min.

### 4.4. Detection of Transgene Excision Efficiency in Greenhouse-Grown Transgenic Plants

Transgenic seedlings were transferred to the greenhouse. At flower stage 9, five to ten flowers from each transgenic plant were bagged for self-pollination. After six weeks, approximately 300–1500 seeds from each plant were collected and germinated at 25 °C. After two weeks of germination, the efficiency of transgene excision was investigated in the fully developed T1 seedlings using GUS staining, following the protocols described above.

### 4.5. Confirmation of Transgene Excision in the T1 Generation of Greenhouse-Grown Transgenic Plants

Additional seeds (at least 20,000 per line) from T1 transgenic lines (showing negative GUS staining) were selected and germinated for further confirmation. Lines producing no blue-stained T1 seedlings during this second round of testing potentially had a transgene excision efficiency of 100%. All subsequent experiments were performed using these transgenic lines.

Confirmed transgenic T1 plants were then self- and cross-pollinated. For cross-pollination, the immature anthers were removed from wild-type and transgenic flowers at stage 9, and mature pollen grains collected from transgenic plants were placed on the stigmas of the emasculated wild-type flowers. Simultaneously, pollen grains collected from wild-type plants were placed on the stigmas of the emasculated transgenic flowers. Seeds produced using self- and cross-pollination were collected separately and germinated. The developed seedlings were GUS-stained as described above.

### 4.6. Estimation of Transgene Copy Numbers in Transgenic Plants

The Gene Deletor copy number in transgenic *N. tabacum* was calculated using qPCR. Serial dilutions were prepared using 0.5 × 10^−4^ ng, 1 × 10^−4^ ng, 2 × 10^−4^ ng, 4 × 10^−4^ ng, and 8 × 10^−4^ ng of ‘Gene Deletor’ plasmid DNA with 30 ng of wild-type *N. tabacum* genomic DNA.

We then measured the relative expression of *FLP* in each mixture using quantitative real-time PCR (qPCR). Primer sequences for pFLP-F and pFLP-R were designed (5′-CAAGAAAACCAGCTGTGACAAGCCTTAAAC-3′ and 5′-CGAGTTCTGCCTCTTTGTGAGTCTCAATAG-3′). Each qPCR reaction (20 μL) contained 2× SoSo Fast EvaGreen Supermix (BioRad, Hercules, CA, USA), 5 μM of each primer, and 30 ng of genomic DNA, with nuclease-free water added to make the final volume. The thermal cycling conditions were as follows: 98 °C for 3 min, followed by 40 cycles of 98 °C for 5 s and 60 °C for 20 s. All qPCRs were performed using a Bio-Rad CFX96/C1000 Real-Time System. The *L25* gene, which has an amplification efficiency approximately equivalent to that of *FLP* [40], was used as an internal reference gene. Next, the standard curve was generated, with *FLP* copy numbers plotted against the ΔCt value. The FLP copy number was calculated using Bio-Rad CFX MANAGER software (http://www.bio.rad.com (accessed on 3 June 2019). Three technical replicates of each experiment were performed for each transgenic line (100% transgene excision efficiency) and control line (0% transgene excision efficiency).

### 4.7. Transgene Excision in Field-Grown Transgenic Plants

Vegetatively propagated greenhouse transgenic lines were cultivated in the field between May and September for three years to determine transgene excision efficiency under field conditions. In total, seven independent field experiments (using subsets of the control and transgene-excised transgenic lines) were conducted over the three years. In each field experiment, the roots, leaves, petioles, stems, flower stalks, and flower organs of each plant at flower stage 1 were stained for GUS activity. The *FLP* gene was also PCR amplified from the leaves of the young plants to test whether the transgenes were present. Transgene excision in the pollen and seeds of the F1 plants was tested using GUS staining, as described above.

We tested the transgene excision efficiency of each control and transgene-excised transgenic line in the field experiments by analyzing GUS activity in seedlings produced via self-pollination and reciprocal crossing. Mature pollen grains and immature seeds were collected from the young fruits of the T1 seedlings at 18 days after pollination (DAP) and GUS stained to test for transgene excision in the pollen and seeds.

### 4.8. Molecular Verification of Transgene Excision in the Transgenic Plants

For transgene excision verification, genomic DNA was isolated from wild-type and transgenic plants in the T_0_ and T1 generations. A primer pair specific to the T-DNA sequences outside the two loxP-FRT sites was designed (pLF-F (5′-TGCAAGGCGATTAAGTTGGGTAAC-3′) and pLF-R (5′-ACCATTATTGCGCGTTCAAAA GTC-3′)). PCRs were performed on a S1000 Thermal Cycler (Bio-RAD, Hercules, CA, USA) with Primestar *Taq* polymerase (TaKaRa, San Jose, CA, USA). Each 25 µL reaction mixture include 30 ng genomic DNA as a template. The PCR cycling conditions were as follows: initial denaturation at 98 °C for 5 min; 10 cycles of touch-down PCR (denaturation at 98 °C for 5 s, annealing at 55–65 °C for 15 s, and extension at 72 °C for 3.5 min); 30 cycles of regular PCR (denaturation at 98 °C for 10 s, annealing at 55 °C, and extension at 72 °C for 2 min), and a final extension at 72 °C for 5 min. Then, 1 µL of PCR product was used as the template for a second PCR with the same primers and the following cycling conditions: initial denaturation at 98 °C for 3 min; followed by 35 cycles of 98 °C for 5 s, 60 °C for 10 s, and 72 °C for 3.5 min; and a final extension at 72 °C for 10 min. The PCR products were resolved using electrophoresis in a 1% agarose gel and photographed using the Bio-Rad Gel Doc system (Bio-RAD, Hercules, CA, USA). About 10–15 PCR amplicons per transgenic line were sequenced to analyze the composition of the amplified DNA fragments. 

## Figures and Tables

**Figure 1 ijms-24-01160-f001:**
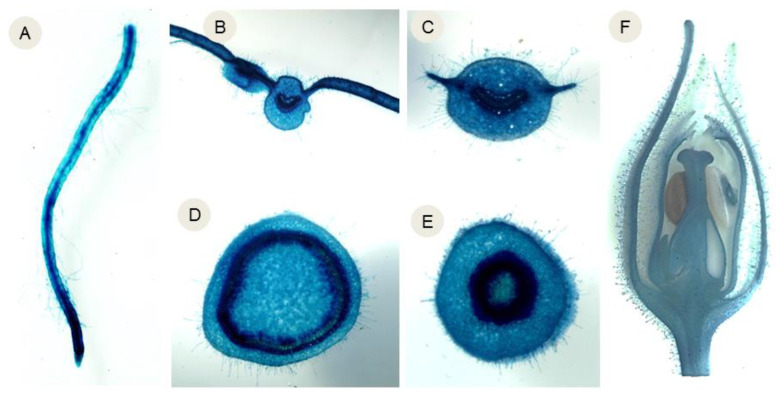
Histochemical analysis showing GUS activity (indicated by blue staining) in various structures of transgenic plants carrying the ‘pollen- and seed-specific Gene Deletor’ cassette. (**A**) Root, (**B**) leaf section, (**C**) petiole section, (**D**) stem section, (**E**) flower stalk section, (**F**) flower section. Scale bar = 20 µM.

**Figure 2 ijms-24-01160-f002:**
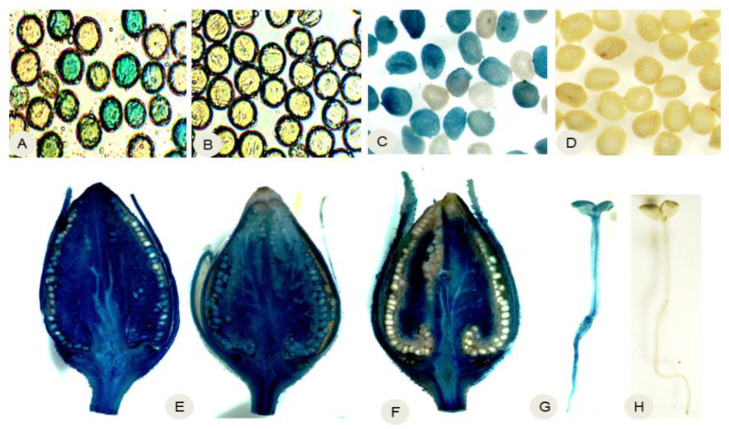
Detection of transgene deletion. (**A**) GUS-stained mature pollen grains produced by control transgenic line C1, showing GUS activity (blue staining) in about half of the pollen grains. (**B**) GUS-stained mature pollen grains produced by transgenic line D4 carrying the ‘pollen- and seed-specific Gene Deletor’ cassette, showing that no pollen grains had GUS activity. (**C**) GUS-stained immature seeds collected from the fruits of line C1 at 18 days after pollination (DAP), showing that some seeds had GUS activity. (**D**) GUS-stained immature seeds collected from the fruits of line D4 at 18 DAP, showing that no seeds had GUS activity. (**E**) GUS-stained sections of an immature fruit produced by line C1 at 18 DAP; many seeds show GUS activity. (**F**) GUS-stained half-section of the immature fruit (18 DAP) produced by line D4, showing white seeds (i.e., without GUS activity) attached to blue tissues (i.e., GUS positive). (**G**,**H**) GUS-stained self-pollinated T1 seedlings produced by (**G**) line C1 and (**H**) line D4. Scale bar = 20 µM.

**Figure 3 ijms-24-01160-f003:**
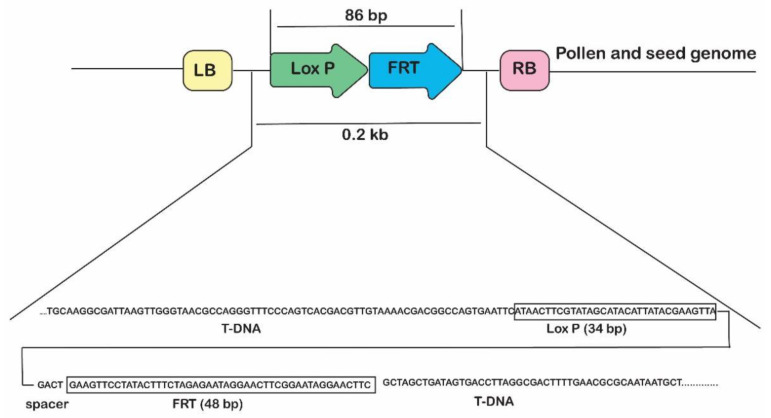
Confirmation of transgene excision in the ‘pollen- and seed-specific Gene Deletor’ system. Schematic representation of the excision of a portion of the transgenic genome. All DNA between the two fused LoxP-FRT sites, including the transgenes, is deleted specifically in the pollen and seeds, and the genomic DNA is then ligated by recombinase FLP, which is encoded by the *FLP* gene under the control of the pollen- and seed-specific promoter, PAB5. After the excision process, a short T-DNA residue (~0.2 kb) containing an intact fused LoxP-FRT site remains in the genome. LB and RB: left- and right-borders of the T-DNA, respectively; LF: fused loxP and FRT (loxP-FRT) recognition sequence; PAB5: pollen- and seed-specific *Arabidopsis* PAB5 gene promoter; P35S: CaMV 35S gene promoter; Tnos: terminator of the *Agrobacterium tumefaciens* nopaline synthase gene; FLP: FLP recombinase gene; NPTII: neomycin phosphotransferase II gene (kanamycin resistance was used as the plant selection marker).

**Table 1 ijms-24-01160-t001:** Numbers of plants transformed with the control or ‘pollen- and seed-specific Gene Deletor’ cassettes and the corresponding transgene excision efficiency in the produced pollen and seeds, based on GUS activity levels in the T1 seedlings grown in the greenhouse.

Transformation	Constructs	No. of Independent Transgenic Lines	Mean Percentage of GUS Negative T1 Seedlings	No. of Lines with 100% Excision Efficiency
Experiment No. 1	Control Cassette	22	19.2%	0
	Gene Deletor	66	37.7%	2
Experiment No. 2	Control Cassette	19	21.4%	0
	Gene Deletor	81	36.1%	2
Experiment No. 3	Control Cassette	21	20.7%	0
	Gene Deletor	55	35.0%	1

**Table 2 ijms-24-01160-t002:** Transgene excision efficiency in the pollen and seeds of transgenic plants grown in a greenhouse; transgene excision efficiency was determined based on GUS staining of the T1 seedlings. Lines C1, C14, and C6 (control lines); lines D4, D10, D31, D56, and D43 (lines with “Gene Deletor” cassette).

Line	Transgene Copies	Self-Pollinated		WT^†^ as Pollen Recipient		WT as Pollen Donor		Transgene Excision Efficiency (%)
		GUS^−^	GUS^+^	GUS^−^	GUS^+^	GUS^−^	GUS^+^	
C1	1	532	1742	789	834	873	951	0.0
D4	1	21,862	0	2654	0	2476	0	100.0
D10	1	41,787	0	3844	0	4366	0	100.0
C14	1	651	1938	745	817	922	893	0.0
D31	1	6874	0	2943	0	3326	0	100.0
D56	1	5796	0	3219	0	3543	0	100.0
C6	1	717	2258	1142	1271	967	1055	0.0
D43	1	5864	0	2227	0	2581	0	100.0

**Table 3 ijms-24-01160-t003:** Transgene excision efficiency in the pollen and seeds of transgenic plants grown in the field; transgene excision efficiency was determined based on GUS staining of the T1 seedlings. Lines C1, C14, and C6 (control lines); lines D4, D10, D31, D56, and D43 (lines with “Gene Deletor” cassette).

Experiment No.	Number of Field Tests	Line	Self-Pollinated		WT as Pollen Recipient		WT as Pollen Donor		Transgene Excision Efficiency (%)
			GUS^−^	GUS^+^	GUS^−^	GUS^+^	GUS^−^	GUS^+^	
Experiment 1	Field Test (1)	C1	658	2046	892	973	795	870	0.0
		D4	49,264	0	6739	0	6025	0	100.0
		D10	53,119	0	6033	0	6989	0	100.0
	Field Test (2)	C 1	506	1647	768	794	691	715	0.0
		D4	44,301	0	5541	0	5905	0	100.0
		D10	64,356	0	7647	0	8036	0	100.0
	Field Test (3)	C1	1628	5034	1773	1826	1698	1735	0.0
		D4	20,449	0	4363	0	4834	0	100.0
		D10	16,841	0	3961	0	3589	0	100.0
Experiment 2	Field Test (4)	C14	1275	3922	1169	1352	1288	1346	0.0
		D31	18,864	0	4032	0	3975	0	100.0
		D56	10,011	0	2763	0	2970	0	100.0
	Field Test (5)	C14	1053	3217	976	1018	1127	1253	0.0
		D31	16,797	0	3524	0	3276	0	100.0
		D56	16,231	0	3226	0	3537	0	100.0
Experiment 3	Field Test (6)	C6	1136	3579	1217	1188	1204	1256	0.0
		D43	14,683	0	2533	0	2369	0	100.0
	Field Test (7)	C6	1317	4183	1479	1514	1396	1447	0.0
		D43	16,068	0	2699	0	2528	0	100.0

**Table 4 ijms-24-01160-t004:** PCR verification of transgene excision efficiency in the T1 seedlings that were produced by self-pollinating transgenic plants in the field. Line C1 (control line) was transformed with the control cassette; lines D4, D10, D31, D56, and D43 were transformed with the ‘pollen- and seed-specific Gene Deletor’ cassette.

Line	PCR Analysis of Self-Pollinated T1 Seedlings				DNA Sequencing Analysis of 0.2 kb Fragments in T1 Seedlings	
	Seedlings Carrying the 7.6 kb Fragment	Seedlings Carrying the 0.2 kb Fragment	Seedlings Not Producing Any Bands	Total Seedlings Tested for PCR	Seedlings Sequenced	Seedlings Carrying Expected 0.2 kb Sequence
C1	21	0	9	30	0	0
D4	0	38	12	50	10	10
D10	0	36	14	50	10	10
D31	0	33	17	50	5	5
D56	0	37	13	50	5	5
D43	0	40	10	50	5	5

## Data Availability

All data generated or analyzed during this study were included in this published article and its Appendix A.

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
