# Peer review of "Confirmation of ‘Pollen- and Seed-Specific Gene Deletor’ System Efficiency for Transgene Excision from Transgenic *Nicotiana tabacum* under Field Conditions"

_ijms, 2023, doi:10.3390/ijms24021160_

Round 1

Reviewer 1 Report (Previous Reviewer 1)

Dear authors,

It is with pleasure that I review your manuscript. All my recommandations have been addressed and I recommand acceptance of your manuscript.

Best regasrds

Author Response

Dear Reviewer.

Thank you so much for your comments and suggestions.

Best Regards.

Wei Yao

Reviewer 2 Report (Previous Reviewer 2)

Comments and Suggestions for Authors

Manuscript ijms-2091536, the authors showed that transgenic plants with 100% transgene-excision efficiency were repeatably produced by transformation with the 'Gene-Deletor' vector, and stable replication replicable under variable climatic conditions over three consecutive years, indicating that the 'Gene-Deletor' system is a reliable automatic transgene-excision system. I suggest that the following minor issues need to be addressed:

Major comments:

1.      Reviewer 2 raised the issue of the lack of novelty in the manuscript and where this part of the improvement is described in the manuscript.

2.      Reviewer 2 showed that the ' Gene-Deletor' system still had some components remaining and that there were no experiments to determine whether these components posed a risk, and where this part of the discussion is reflected in the manuscript.

3.     “PCR analysis showed that the FLP gene was present in the genomes of all transgene-excised transgenic lines tested (lines D4, D10, D31, D56, and D43) but not in control lines (data not shown)”, the expression of Cre sometimes does not truly reflect the efficiency of Cre on Loxp cleavage, and PCR can compensate for this. It is suggested that some of this be added to the appendix of the manuscript to add credibility.

Minor comments:

1.      The figures for T0 and T1 should be subscripted and are inconsistent in the manuscript.

Author Response

Dear Reviewer.

We really appreciate your comments and suggestions. Please see the attachment.

Best Regards.

Wei Yao

This manuscript is a resubmission of an earlier submission. The following is a list of the peer review reports and author responses from that submission.

Round 1

Reviewer 1 Report

Dear authors,

Manuscript ijms-1922370 entiteled "Field evaluation of the ‘pollen- and seed-specific Gene-Deletor’ system with 100% efficiency in tobacco plants" and authored by Zhenzhen Duan , Mingyang He , Sehrish Akbar , Zhao de Gang , Muqing Zhang , Yi Li and Wei Yao targets a hot topic that fits well with the scope of the journal and that is potentially of high interest for5 the journal readers. While the work seems to be nicely designed and the experiments accurately conducted I could not recommend at this stage the manuscript for publication. In fact few points needs authors attention anmd needs to be addressed:

1. The title does not really encourage the readers to read the manuscript. Please change it by stressing more your findings.

2. Introduction section: Please describe more widely the site specific recombinase-mediated transgenic excision. This is the most important part ane non specialized reasers need really details to understand and follow your manuscript.

3. Results section: please improve the readability of your figures actually figure 3 is difficult to read.

4. you clain in the discussion section lines 252-254 "After transgene excision, only a very short foreign DNA fragment, containing the left- and right- boarder T-DNA sequences, as well as an intact loxP-FRT recognition sequence, remained in the genomes of the pollen and seeds produced by the transgenic plants" what about the effect of this part on the genome evolution of the species? it is external DNA anyway so please discuss this part and convince readers why they should not be inquired.

I am really looking forward to read an improved version of this manuscript that I could recommend for publication.

Best regards

Author Response

Please see the attachment, Thank you so much for your comments and suggestions.

Reviewer 2 Report

The authors conducted field experiments over three consecutive years to evaluate the stability of transgene excision under field conditions. The result may provide some useful information for the food safety.

However, the basic problem for the manuscript is the lack of novelty. Thus, this study is not sufficient to merit publication in International Journal of Molecular Sciences.

1. The vector system used to excise transgene in tobacco has been reported previously (Luo et al. 2007, DOI: 10.1111/j.1467-7652.2006.00237.x), only the environmental factors were changed, which is lack of novelty.

2. In transgenic plants, the transgenic elements may recombine or disrupt, when the elements do not work, false negatives will not be distinguished by GUS histochemical staining. In this case, this system can’t excise the foreign elements, and is of little significance.

3. The authors highlighted that the ‘Gene-Deletor’ system excise transgene and confer no risk, however, there are still components remaining, and there are no experiments to determine whether these components bring no risks.

4. There are figures missing in the manuscript, for example, figure 4 is not found in the text or attachments.

5. The consecutive experiment time does not consist throughout the text, for example, line 84, “we tested the stability of transgene excision in the field over four consecutive years”, but in the abstract, line 361 and so on, “three years” was described.

6. There are some problems in the manuscript format. For example, the Abbreviation of PLANT BIOTECHNOLOGY JOURNAL is PLANT BIOTECHNOL J., not Plant biotechnol. Please check the whole manuscript carefully.

Author Response

(The authors gave the same response as above.)

Round 2

Reviewer 1 Report

Dear authors,

Your manuscript meets now the journal standards and I could suggest it for publication.

Best regards